# Reason with Thumbnails, Answer with Focus: An Efficient and Effective Paradigm for Multimodal Grounded Visual Reasoning

An-Lan Wang [1]  Guozhi Tang [2]  Lei Liao [2]  Hanshen Zhu [2]  Kai Huang [2]  Jingqun Tang [2]  Jiaming Zhou [3]
Kun-Yu Lin [4]

## Abstract

To enhance the interpretability of multimodal large language models' outputs, recent efforts explored Grounded Visual Reasoning (GVR), in which the model is trained to select relevant image regions before answering the question. However, the multi-round "ground-then-answer" and reasoning nature of these methods impose much more computational costs compared to non-GVR methods. To attain efficient and effective GVR, in this paper, we propose a novel paradigm called **R**eason with **T**humbnails, **A**nswer with **F**ocus (RTAF), which feeds the model with low-resolution images to reason the key area and high-resolution crops to answer the final answer. Our motivation arises from the observation that, in many cases, the key area required to answer questions can be inferred from the low-resolution thumbnails, without the need for a full-resolution image. Additionally, for extreme cases where thumbnails lack sufficient information (leading to undirected region guessing), we equip the model with a tool to access higher-resolution images. For training efficiency, we adopt pure reinforcement learning and design a suite of reward functions to supervise the model's behavior, alongside a resolution-aware training data selection strategy. Finally, our model, based on Qwen2.5-VL, achieves significant improvements across a range of benchmarks with reduced computation, demonstrating the effectiveness and efficiency of our proposed RTAF, *e.g.*, compared to Qwen2.5-VL, our model achieves a performance gain of 5.8% while using a comparable number of visual to-

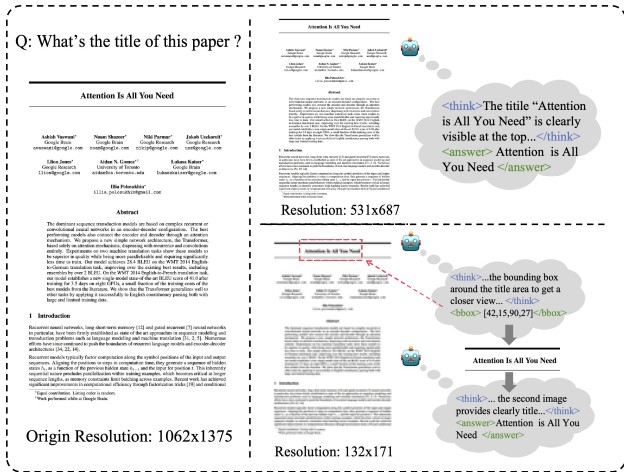

Figure 1. Illustration of the motivation for "*Reason with Thumbnails, Answer with Focus*". Given the question and thumbnails with a resolution of 1/4 of the original, the model can directly answer the question as the title is still clearly visible at the top of the image. More extremely, as shown in the example on the bottom right, using a thumbnail with a resolution reduced to 1/64 of the original, it is not challenging to locate the bbox of the title.

kens. Against state-of-the-art GVR methods, our model reduces visual token usage by half while delivering superior performance.

## 1. Introduction

In recent years, we have witnessed remarkable progress in Multimodal Large Language Models (MLLMs) (Bai et al., 2025; Achiam et al., 2023; Wang et al., 2025a; Team et al., 2023; Chen et al., 2023b; Li et al., 2023) in a wide range of real-world tasks, *e.g.,* VQA (Mathew et al., 2021; Wang et al., 2025b; Li et al., 2024; Liu et al., 2026), grounding (Kazemzadeh et al., 2014; Xia et al., 2025), captioning (Lin et al., 2025). Despite their success, conventional MLLMs typically rely on a "direct answer" approach to generate the final answer without explicit reasoning, which suffers from low answer reliability and poor interpretability. Additionally, such an approach makes the model prone to hallucinations.

[1]School of Computer Science and Engineering, Sun Yat-sen University, China [2]Independent Researcher [3]Hong Kong University of Science and Technology (Guangzhou) [4]The University of Hong Kong. Correspondence to: Kun-Yu Lin <kunyulin14@outlook.com>.

*Proceedings of the $43^{rd}$ International Conference on Machine Learning*, Seoul, South Korea. PMLR 306, 2026. Copyright 2026 by the author(s).

To enhance the interpretability of MLLMs' output, Grounded Visual Reasoning (GVR) (Cao et al., 2025; Zheng et al., 2025) is explored, in which the model is trained to select relevant image regions before answering the question during the chain-of-thought process. However, the multi-round, *i.e.,* "ground-then-answer" and reasoning nature of these methods imposes much more computational costs compared to traditional non-GVR methods. Specifically, for every rollout round, the context, including the original image, textual query, and evidence-aligned image regions, is fed to the model to generate further responses.

To address these issues, we propose a novel GVR paradigm, called Reason with Thumbnails, Answer with Focus (RTAF). Our motivation stems from a scenario that probably everyone has encountered: due to network latency, when loading an image, only a thumbnail (low-resolution image) is displayed first. However, we can already grasp most of the content based on the context and the thumbnail. In this case, when the full image finishes loading, our attention focuses directly on the key regions. That is to say, we can obtain a significant amount of useful information from a thumbnail (low-resolution image), such as the general content and the main subject of the image, which can already support the grounded visual reasoning process. Two examples are provided in Figure 1.

Specifically, unlike previous methods that use the full-resolution original image for reasoning, we input a low-resolution image to the model and let it reason the key region for solving the problem. Subsequently, through coordinate transformation, we crop from the full-resolution original image and provide this cropped image to the model for further reasoning. This process can be executed in multiple rounds until the model locates the correct key regions and answers the question in a specified format, *i.e.,* "`<answer>final answer here</answer>`". Additionally, in some more extreme scenarios where the model deems that no information (e.g., the key region related to the question) can be obtained from this low-resolution image. This leads the model to perform unguided guessing of the key region, ultimately incurring higher computational costs. To address this issue, we further enable the model to acquire an image with higher resolution through a special "`</Resolution>`" tag. The model will then receive an image with higher resolution for further reasoning and answering.

However, our experiments reveal that direct training under this new paradigm leads to performance degradation, primarily stemming from the increased difficulty the model faces when conducting reasoning directly with low-resolution inputs. To mitigate this issue, we further design a suite of reward functions to supervise the model's behavior, alongside a resolution-aware training data selection strategy that filters for high-value training data to facilitate effective training.

To evaluate the effectiveness of our proposed RTAF, following previous works (Shao et al., 2024; Cao et al., 2025), we select a comprehensive suite of benchmarks for evaluation, including visual question answering (Shao et al., 2024), visual grounding (Kazemzadeh et al., 2014). Compared with previous works, our method use fewer visual tokens and achieves better or comparable performance. Furthermore, ablations and case studies verify the proposed paradigm, RTAF, and the training strategies.

Overall, our contributions can be summarized as follows:

- Firstly, we propose a novel GVR paradigm, called Reason with Thumbnails, Answer with Focus (RTAF), which feeds the model with low-resolution images to reason the relevant regions and high-resolution crops to generate the final answer.
- Secondly, we design a suite of reward functions to supervise the model's behavior, including a reward designed specifically for reasoning efficiency. Additionally, we propose a resolution-aware training data selection strategy for effective training.
- Thirdly, extensive experiments and ablations demonstrate the effectiveness of our method. Compared with previous models (both non-GVR and GVR), our method shows effectiveness and efficiency.

## 2. Related Works

### 2.1. Grounded Visual Reasoning

Recent grounded visual reasoning works propose to selectively identify crucial regions to facilitate interpretable reasoning, particularly in complex and visually cluttered scenarios, which is essential for real-world applications (Wu et al., 2026; Zhou et al., 2025a;b; Wei et al., 2024; 2025; Jiang et al., 2025). Early approaches introduce bounding boxes into chain-of-thought (CoT) (Zhang et al., 2023) reasoning via prompting (Mitra et al., 2024) or supervised fine-tuning (SFT) (Chen et al., 2023a). In addition, tool-based methods (Zhou et al., 2024; Lei et al., 2025; Yang et al., 2023) provide tools like cropping and zooming, enabling explicit grounding. More recent works (Sarch et al., 2025; Fan et al., 2025; Cao et al., 2025; Zheng et al., 2025) use reinforcement learning for grounded reasoning, and employ multi-round grounded roll-out for better interpretability. While these grounding-enhanced methods demonstrate impressive performance and improved interpretability, the multi-round and reasoning nature of these methods imposes higher computational overhead compared to traditional non-GVR (Bai et al., 2025) methods. To address these issues, in this work, we propose a new paradigm, Reason with Thumbnails, Answer with Focus (RTAF), in which we input a low-resolution image to the model for reasoning the key region, while using the high-resolution crops for answering. Ultimately, our

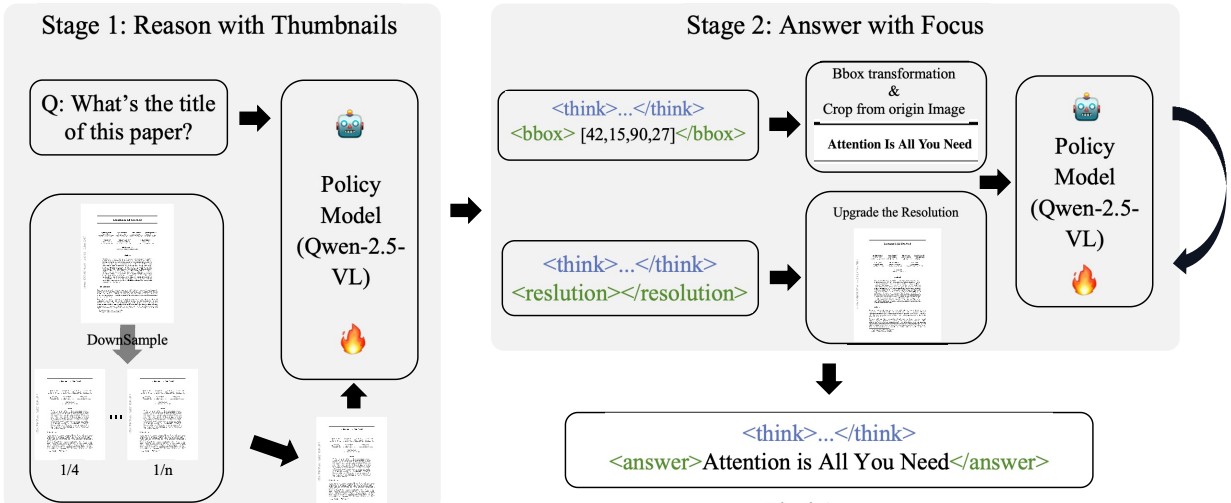

*Figure 2.* Pipeline of our proposed Reason with Thumbnails, Answer with Focus (RTAF) framework: In Stage 1, the model reasons over low-resolution thumbnails (downsampled from the original image) to identify relevant regions. It either outputs the bounding boxes to obtain a high-resolution image crop or attains an image with higher resolution, with this iterative process repeated for multiple rounds. Then, based on all previous context, the final answer is generated.

method achieves both effective and efficient GVR.

## 2.2. RL with Verifiable Rewards

Reinforcement Learning with Verifiable Rewards (RLVR) has been explored as a powerful tool for improving the generalizability and interpretability of MLLMs. A pioneering work, DeepSeek-R1 (Guo et al., 2025), demonstrates that pure reinforcement learning with simple verifiable rewards (*i.e.,* format and accuracy rewards) can stimulate active reasoning processes in large language models, fostering the emergence of abilities like self-reflection and self-correction. The rewards (Wang et al., 2024a) play a crucial role in this process, which directly assigns an overall score to the model response. Some studies (Wang et al., 2024b; Balashankar et al., 2024) expanded the reward signals tailored to the desired dimensions, *e.g.,* helpfulness, correctness, coherence, complexity, and verbosity. To mitigate the reward hacking, reward ensembling (Coste et al., 2023) is proposed, which incorporates multiple rewards to reduce the chance of a flat objective landscape. In this work, our proposed model incorporates the image into the reasoning process, equipped with image operation tools like cropping and upsampling. Guided by a set of carefully designed rewards, the model performs iterative visual reasoning efficiently and effectively, trained purely via reinforcement learning.

## 3. Methods

To empower MLLMs with an efficient and effective Multimodal Chain-of-Thought capabilities, we propose **R**eason with **T**humbnails, **A**nswer with **F**ocus (RTAF).

### 3.1. Stage I: Reason with Thumbnails

Given the original image $i_{org}$ with a resolution of $H \times W$ and a corresponding question $q$, we first generate $n-1$ thumbnails $i_{thumb} = [i_{thumb}^2, ..., i_{thumb}^n]$ through downsampling, where the $k-$th, $k \in [2, n]$, thumbnail has a resolution $H/k \times W/k$. Then, we feed the model with the thumbnail $i_{thumb}^n$ along with the question and prompt.

As shown in the prompt in Figure 3, we formally instruct the model that the input image is a downsampled thumbnail from the original image. The model is then guided to reason, based on this low-resolution thumbnail and the associated question, whether there exists a specific region within the image that can better assist in answering the question. If such a region is identified, the model should provide a bounding box coordinates in the format [x1,y1,x2,y2] enclosed within the "`<box>`" and "`</box>`" tags. Upon providing this coordinate, a cropped image (from the original high-resolution image using the specified bounding box) will be returned.

Additionally, we consider extreme cases where the model fails to acquire any valid information and thus randomly guesses the key region without direction. This random guessing not only increases computational costs but also leads to lower accuracy. Therefore, we further equip the model with a tool to get images with higher resolution, *i.e.,* the model can request an image with higher resolution using a "`</Resolution>`" tag, after which a higher-resolution version will be returned. This process can be executed in multiple rounds until the model locates the correct key region.

---

**Prompt**

Question: {Question}
Based on the low-resolution image (dowmsampled from high-resolution image) and the question, reason whether there exists a region in the image that could help you answer the question better. If such a region exists, provide one bounding box coordinate in the format [x1,y1,x2,y2] inside the \<box\> and \</box\> tags. Then, you will receive a cropped image (cropped from the high-resolution original image using the transformed bounding box).Use all images provided to continue reasoning inside a new \<think\> tag. You may conduct multiple rounds of grounding to refine your region as you want. Do not provide the same bounding box multiple times.
If you find the low-resolution image is insufficient for locating the region that could help you answer the question, you may ask for an image with higher resolution through a \<resolution\> \</resolution\> tag. Then you will receive an image with higher resolution.
If at any point you determine no further visual information is needed, you may DIRECTLY provide the final answer inside the \<answer\> and \</answer\> tags.
*Format Example: \<think\> Reasoning \</think\> \<box\>[x1,y1,x2,y2]\</box\> OR \<think\> Reasoning \</think\> \<resolution\> \</resolution\> OR \<think\> Reasoning \</think\> \<answer\> final answer \</answer\>.*

*Figure 3.* Prompt for RTAF.

### 3.2. Stage II: Answer with Focus

After all preceding "*Reason with Thumbnails*" steps, if the model determines that no additional bounding boxes or higher-resolution images are needed, this indicates that the relevant key regions identified through previous roll-out rounds have provided sufficient information to solve the problem. In such cases, the model directly outputs the final answer within the "`<answer>final answer here</answer>`" tag, namely "*answer with focus*", which is formalized as:

$$o \sim \pi_{\theta_{\text{old}}}(\cdot \mid q, v, e), \tag{1}$$

where $e$ denotes the context information (*e.g.,* preceding reasoning trace) that supplements the query $q$ and visual input $v$ to guide the old model $\pi_{\theta_{\text{old}}}$ in generating the final answer $o$.

### 3.3. Reward Functions

To supervise the model's behavior, we design a suite of rule-based verifiable reward functions.

**Format Reward.** Following previous works (Guo et al., 2025; Cao et al., 2025), we introduce a format reward $R_{format}$. The format reward $R_{format}$ is used to ensure the model's prediction meets the required output format, which is illustrated in the previous sections. This format encourages the model to perform reasoning explicitly before tool usage and answering questions, improving the interpretability.

**Accuracy Reward.** For questions that have amenable numerical verifiable answers, a binary accuracy reward is adopted, which performs an exact answer match against the ground-truth answers. For the free-form questions, following previous works (Guo et al., 2025), the reward is defined as the lexical alignment using the average of ROUGE-1,

ROUGE-2, and ROUGE-L scores against the ground-truth answers. Notably, if a response reaches the maximum number of turns or exceeds the context length limit, the accuracy reward will be set to 0, since no valid answer can be generated under such circumstances.

**Bbox Reward.** To evaluate the model's accuracy in locating answer-related regions via bounding boxes (bbox), a bbox reward is designed based on the overlap between the model-generated bbox and the ground-truth bbox. Specifically, the Intersection over Union (IoU) metric is adopted to quantify the spatial alignment: if the IoU of the generated bbox and ground-truth bbox is greater than or equal to a predefined threshold (*e.g.*, 0.5), the bbox reward is set to 1 (indicating accurate localization); otherwise, it is set to 0. This reward aims to incentivize the model to generate precise, question-relevant bboxes.

**Efficiency Reward.** To encourage the model to answer the question directly when the answer is clearly visible, we add an efficiency reward. Specifically, when no tool is used in the answer rollout process, and the model answers the question correctly, the efficiency reward is set to 1, otherwise it is set to 0. This reward is intended to enforce the model not to use tools ***only*** when the questions are straightforward enough to answer even with just a thumbnail. Formally, the Efficiency reward $R_{eff}$ is defined as:

$$R_{eff} = \begin{cases} 1, & \text{if } R_{acc} = 1 \text{ and no tool used,} \\ 0, & \text{otherwise.} \end{cases} \tag{2}$$

Overall, the total reward $R$ consists of four parts, including the Format Reward, Accuracy Reward, Bbox Reward, and Efficiency Reward:

$$R = R_{format} + R_{acc} + R_{bbox} + R_{eff}. \tag{3}$$

Integrating the Efficiency Reward also offers an advantage: the model will consistently receive the same total reward,

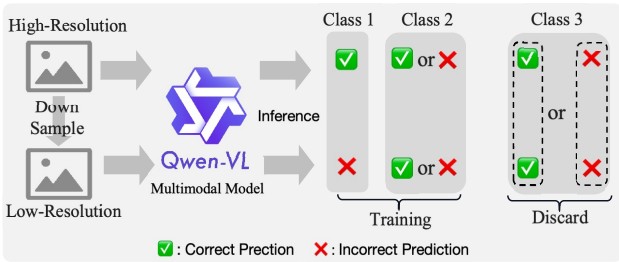

*Figure 4.* Illustration of our Resolution-Aware Training Data Selection Pipeline. For each training sample, we generate both high-resolution and downsampled low-resolution versions, then perform inference using a multimodal model (*i.e.*, Qwen2.5-VL 7B) multiple times. Samples are categorized into three classes based on the inference results: Class 1 (resolution sensitive, correct only in high resolution), Class 2 (resolution-neutral, similar in high/low res), and Class 3 (discard samples, overly difficult or overly simple data). Only Class 1 and Class 2 samples are used for model training.

whether any tool is used, when the answer is correct, emphasizing that answering the question correctly is the primary goal.

### 3.4. Resolution-aware Training Data Selection

During the training process of GRPO, overly difficult or overly simple data can easily lead to training instability and inefficiency, such as reward collapse (Coste et al., 2023). To enhance training stability and efficiency, we propose a Resolution-aware Training Data Selection pipeline, as shown in Figure 4.

Specifically, for all original training data, we generate a low-resolution version and perform multiple inferences (*i.e.,* 8 times) using Qwen2.5-VL. Then, based on the inference results, the original data are categorized into three classes:

- Class 1: Resolution Sensitive (relying on high-resolution details for the correct answer, only predicted correctly when fed with high-resolution images, and failing in low-resolution inference)
- Class 2: Resolution-Neutral (similar results across high and low resolutions)
- Class 3: Discard Samples (overly difficult/simple data, always correct or always incorrect in both resolutions)

For our model, Class 1 has the highest value, so we incorporate all available samples from Class 1 into the training dataset, while Class 3, with the lowest value, is discarded.

## 4. Experiments

### 4.1. Implementation Details

The policy model in the main paper is Qwen2.5-VL-Instruct. Following previous work (Cao et al., 2025), the rollout numbers for the "*reason with thumbnails*" and "*answer with focus*" stages were set to 4 and 2, respectively, and the sampling temperature is set to 1; the maximum response length is set to 512 tokens. For the input resolution, we adopt a resolution-adaptive strategy: if the image resolution exceeds 4096 in either dimension, we use a 1/16 resolution scale; if it falls between 1024 and 4096, a 1/9 scale is applied; otherwise, a 1/4 scale is used. For the final training dataset, we use all Class 1 samples (2k samples) and 6k samples from Class 2 (randomly sampled), resulting in a total of 8k samples for 1,000 training steps across 8 GPUs.

The model is trained with GRPO (Guo et al., 2025) for 1000 steps with a batch size of 8 and a learning rate of 1e-6. Following previous works (Guo et al., 2025; Cao et al., 2025), a KL divergence regularization is adopted to prevent excessive policy drift from the reference model, and the regularization coefficient $\beta$ is set to 0.04. The maximum number of turns is set to 4; otherwise, the reasoning sequence will be truncated.

### 4.2. Benchmarks and metrics

Following previous works, to systematically evaluate our proposed RTAF, we first use a suite of benchmarks from VisCoT, which specifically focuses on identifying critical regions within images, including DocVQA (Mathew et al., 2021), TextCaps (Sidorov et al., 2020), TextVQA (Singh et al., 2019), DUDE(Van Landeghem et al., 2023), SROIE (Huang et al., 2019), InfoVQA (Mathew et al., 2022), Flickr30k (Plummer et al., 2015), Visual7W (Zhu et al., 2016), GQA (Hudson & Manning, 2019), Open Images (Kuznetsova et al., 2020), VSR (Liu et al., 2023), and CUB (Wah et al., 2011). Additionally, to validate the grounding ability of our proposed method, we further evaluate the model using the RefCOCO (Kazemzadeh et al., 2014) and RefCOCO+ (Kazemzadeh et al., 2014).

For the VisCoT benchmark, we report both the accuracy and average number of visual tokens to enable a comprehensive comparison in terms of effectiveness and efficiency. For the grounding benchmarks, we report the evaluation results in testA, testB, and val.

### 4.3. Main Results

As shown in Table 1, we compare our proposed method with two types of methods, i.e., GVR methods and non-GVR methods, and we report the accuracy on these benchmarks as well as the average visual tokens used. Compared with previous non-GVR methods, our method achieves significant improvements. Specifically, it surpasses LLaVA-1.5-13B by +30.5% in average accuracy, and outperforms Qwen2.5-VL-7B by +5.8% while using a comparable number of visual tokens (471 *vs.* 391). Compared with previous GVR methods like Ground-R1, our method achieves better accuracy

*Table 1.* **Evaluation results on the test split of VisCoT (Shao et al., 2024) benchmark** including doc/text and chart understanding and general VQA, relation reasoning, and fine-grained VQA (FGVQA). "AVG V Token" indicates the Average Visual Token used per image in the benchmarks. [†] To calculate the Average Visual Token number, we reproduce the Ground-R1 following the official instructions and add the bbox reward with randomly sampled training data for comparison.

| Method | AVG V Token | AVG | Doc/Text | | | | | Chart |
|---|---|---|---|---|---|---|---|---|
| | | | DocVQA | TextCaps | TextVQA | DUDE | SROIE | InfoVQA |
| LLaVA-1.5-7B (Liu et al., 2024b) | 576 | 45.4 | 24.4 | 59.7 | 58.8 | 29.0 | 13.6 | 40.0 |
| LLaVA-1.5-13B (Liu et al., 2024b) | 576 | 47.8 | 26.8 | 61.5 | 61.7 | 28.7 | 16.4 | 42.6 |
| SPHINX-13B (Liu et al., 2024a) | - | 41.9 | 19.8 | 55.1 | 53.2 | 0.0 | 7.1 | 35.2 |
| VisCoT-7B (Shao et al., 2024) | - | 58.0 | 47.6 | 67.5 | 77.5 | 38.6 | 47.0 | 32.4 |
| CogCoM (Qi et al., 2024) | - | – | – | – | 71.1 | – | – | – |
| Chain of Spot (Liu et al., 2024c) | - | – | – | – | 60.9 | – | – | – |
| FAST (Sun et al., 2024) | - | – | – | – | 60.7 | – | – | – |
| Vision-R1-7B (Huang et al., 2025) | - | 73.1 | 86.0 | 71.9 | 88.3 | 67.0 | 89.9 | 61.3 |
| LMM-R1 (Peng et al., 2025) | - | 69.2 | 84.3 | 78.0 | 91.2 | 66.7 | 78.5 | 55.6 |
| R1-Onevision (Yang et al., 2025) | - | 73.5 | 81.0 | 75.0 | 86.3 | 67.7 | 83.7 | 56.7 |
| Ground-R1[†] (Cao et al., 2025) | 847 | 77.8 | 90.9 | 80.7 | **94.2** | 67.3 | 89.2 | **68.1** |
| Qwen2.5-VL-7B (Bai et al., 2025) | 391 | 72.5 | 88.0 | **81.5** | 88.4 | 72.3 | 88.7 | 67.1 |
| Ours | 471 | **78.3** | **94.7** | 79.5 | 93.7 | **76.9** | **93.7** | 65.9 |

| Method | AVG V Token | AVG | General VQA | | Relation Reasoning | | | FGVQA |
|---|---|---|---|---|---|---|---|---|
| | | | Flickr30k | Visual7W | GQA | Open images | VSR | CUB |
| LLaVA-1.5-7B (Liu et al., 2024b) | 576 | 45.4 | 58.1 | 57.5 | 53.4 | 41.2 | 57.2 | 53.0 |
| LLaVA-1.5-13B (Liu et al., 2024b) | 576 | 47.8 | 62.0 | 58.0 | 57.1 | 41.3 | 59.0 | 57.3 |
| SPHINX-13B (Liu et al., 2024a) | - | 41.9 | 60.7 | 55.8 | 58.4 | 46.7 | 61.3 | 50.5 |
| VisCoT-7B (Shao et al., 2024) | - | 58.0 | 66.8 | 55.8 | 63.1 | 82.2 | 61.4 | 55.9 |
| CogCoM (Qi et al., 2024) | - | – | – | – | 71.7 | – | – | – |
| Chain of Spot (Liu et al., 2024c) | - | – | – | – | 63.7 | – | – | – |
| FAST (Sun et al., 2024) | - | – | – | – | 63.8 | – | – | – |
| Vision-R1-7B (Huang et al., 2025) | - | 73.1 | 64.4 | 47.2 | 79.0 | 74.0 | 67.0 | 81.3 |
| LMM-R1 (Peng et al., 2025) | - | 69.2 | 61.7 | 52.6 | 80.0 | 76.7 | 40.2 | 64.8 |
| R1-Onevision (Yang et al., 2025) | - | 73.5 | 68.6 | 68.8 | 75.0 | 72.0 | 73.0 | 74.0 |
| Ground-R1[†] (Cao et al., 2025) | 847 | 77.8 | 60.4 | 61.9 | 89.3 | 82.7 | **71.0** | 82.7 |
| Qwen2.5-VL-7B (Bai et al., 2025) | 391 | 72.5 | **64.3** | 50.7 | 81.7 | 78.0 | 57.1 | 52.4 |
| Ours | 471 | **78.3** | 55.3 | **62.2** | **93.7** | **87.4** | 67.3 | **82.7** |

*Table 2.* Evaluation results of grounding ability on RefCOCO and RefCOCO+.

| Model | RefCOCO | | | RefCOCO+ | | |
|---|---|---|---|---|---|---|
| | testA | testB | val | testA | testB | val |
| Ground-R1 | 93.9 | 88.0 | 92.9 | 90.8 | 78.8 | 86.5 |
| Qwen2.5-VL-7B | 92.5 | 85.4 | 90.0 | 89.1 | 76.9 | 84.2 |
| Ours | **96.3** | **89.1** | **93.3** | **94.3** | **83.4** | **89.4** |

*Table 3.* Evaluation results on high-resolution benchmarks (HR-Bench-4K and HR-Bench-8K).

| Method | HR-4K | | HR-8K | |
|---|---|---|---|---|
| | AVG V Token ↓ | ACC ↑ | AVG V Token ↓ | ACC ↑ |
| Ours | **13118** | 73.8 | 26772 | **69.5** |
| Qwen2.5VL-7B | 15829 | 68.2 | **16269** | 62.7 |
| DeepEyes | 41080 | 73.2 | 30999 | **69.5** |
| Pixel Reasoner | 39418 | **74.0** | 35856 | 66.9 |

(78.3 *vs.* 77.8) with a substantial reduction in average visual tokens (471 *vs.* 847). Specifically, our method achieves SOTA results in 7 out of 12 benchmarks, e.g., an improvement of +3.8% on DocVQA and +4.4% in GQA. The results demonstrate that our method effectively answers the questions correctly with slightly more visual tokens, which is attributed to the proposed RTAF paradigm.

Additionally, we evaluate grounding ability on the RefCOCO (Kazemzadeh et al., 2014) and Ref-COCO+ (Kazemzadeh et al., 2014), where the model is tasked with outputting the bounding box of a specified object in a single-turn interaction; the results are shown in Table 2. Our model significantly outperforms Ground-R1 and Qwen2.5-VL-7B across all splits: on RefCOCO testA, it reaches 96.3 (*vs.* 93.9 for Ground-R1 and 92.5 for Qwen2.5-VL-7B); on RefCOCO+ testB, it achieves 83.4 (*vs.* 78.8 and 76.9, respectively). These results collectively validate that our method enhances grounding precision across diverse benchmarks, showcasing robust region-level reasoning capabilities.

Furthermore, we evaluate RTAF on two high-resolution benchmarks (HR-Bench-4K (Wang et al., 2025d) and HR-

*Table 4.* Ablation study of our method on the VisCoT benchmark in terms of Accuracy (ACC) and Average Visual Token (AVG V Token) used.

| Model | RTAF | Rewards | Data Selection | AVG V Token ↓ | ACC↑ |
|---|---|---|---|---|---|
| Baseline | | | | 847 | 77.8 |
| + RTAF | ✓ | | | 627 | 76.2 |
| + Reward | ✓ | ✓ | | 553 | 77.1 |
| + Data | ✓ | ✓ | ✓ | 471 | 78.3 |

*Table 5.* Effect of the reward functions on the VisCoT benchmark in terms of Accuracy (ACC), Average Visual Token (AVG V Token), and Average Trajectory Length (AVG Traj. Len.).

| Rewards | AVG V Token ↓ | ACC ↑ | AVG Traj. Len. |
|---|---|---|---|
| $R_{format} + R_{acc}$ | 627 | 76.2 | 3.2 |
| $R_{format} + R_{acc} + R_{bbox}$ | 592 | 76.7 | 3.0 |
| $R_{format} + R_{acc} + R_{bbox} + R_{eff}$ | 553 | **77.1** | 2.5 |
| $R_{format} + R_{acc} + R_{bbox} + \mathbf{2 \times R_{eff}}$ | **282** | 74.5 | **1.4** |

*Table 6.* Effect of the training data selection strategy on the VisCoT benchmark in terms of Accuracy (ACC) and Average Visual Token (AVG V Token) used.

| Training Data Selection | AVG V Token ↓ | ACC↑ |
|---|---|---|
| Random | 553 | 77.1 |
| Combination 1 | 551 | 77.6 |
| Combination 2 | 513 | 77.6 |
| Combination 3 | 471 | 78.3 |

Bench-8K (Wang et al., 2025d)) and compare it with Pixel Reasoner (Wang et al., 2025c) and DeepEyes (Zheng et al., 2025). The results are as follows in Table 3. On the HR-Bench-4K benchmark, our method uses the lowest average number of visual tokens (13,118) compared to all baselines, saving over 66% of tokens compared to DeepEyes (41,080) and Pixel Reasoner (39,418). Despite this massive reduction in token usage, our method maintains a highly competitive accuracy (73.8), outperforming DeepEyes (73.2) and Qwen2.5VL-7B (68.2) and matching closely with Pixel Reasoner (74.0). On the HR-Bench-8K benchmark, our method achieves the highest accuracy (69.5, tied with Deep-Eyes) while consuming significantly fewer tokens (26,772 *vs.* 30,999). Although Qwen2.5VL-7B uses fewer tokens in the 8K setting, its accuracy drops substantially to 62.7. These results collectively demonstrate that the proposed RTAF paradigm achieves an optimal balance between high-resolution performance and computational efficiency, significantly reducing token consumption without sacrificing reasoning accuracy on complex high-resolution tasks.

## 4.4. Ablation Study

**Effects of the main components.** In Table 4, we analyze the effect of each component of our proposed method. For the baseline, we apply the Ground-R1 (Cao et al., 2025), using the format and accuracy reward functions. By adopting the RTAF paradigm, the average visual token drops to 627 (a decrease from 847 of the baseline), though the accuracy slightly drops to 76.2. To account for this, we attribute it primarily to the increased difficulty for the model when directly performing reasoning with low-resolution inputs. Then, by introducing new rewards, our model obtains a lower average visual token of 553 and an accuracy of 77.1, *i.e.*, the rewards

help improve accuracy while further reducing token usage, which demonstrates the effectiveness of the reward design in balancing efficiency and performance. Then, by introducing the data selection strategy, the average visual token is further reduced to 471, and the accuracy reaches 78.3, which demonstrates the effectiveness of data selection in optimizing both efficiency and accuracy. Overall, if we observe the trend across all components, each additional component contributes to reducing token usage and, in most cases, improving accuracy, indicating the cumulative benefits of our proposed method's components.

**Effects of the reward functions.** In Table 5, we ablate the items in the overall reward functions and track Average Trajectory Length (which acts as a proxy for tool usage, where shorter trajectories indicate reluctance to use tools). The baseline uses the format and accuracy rewards. First, introducing the bounding box reward ($R_{bbox}$) reduces the average visual token count from 627 to 592 while boosting accuracy to 76.7, indicating its role in refining region localization. Subsequently, incorporating the calibrated efficiency reward ($R_{eff}$) leads to a further reduction in average visual tokens to 553, alongside the highest accuracy of 77.1. At this setting, the model maintains a healthy average trajectory length of 2.5, proving it actively and willingly uses the tool for necessary visual exploration.

To empirically demonstrate that our calibrated efficiency reward does not make the model overly cautious in practice, we add an experiment with a doubled efficiency penalty ($2 \times R_{eff}$). As shown in the last row, when we overweight the efficiency reward, the trajectory length drastically drops to 1.4, which proves that an overweighted efficiency reward forces the model to become "overly cautious" and answer directly from the thumbnail without sufficient exploration, which consequently hurts the accuracy (dropping to 74.5). These results underscore that our proposed calibration of $R_{eff}$ is sufficient and optimal in practice, effectively lowering computational overhead without compromising the model's reasoning capabilities.

**Analysis of the training data.** In our method, we introduce a new training data selection pipeline for effective training. For a thorough analysis, in Table 6, we report the results of using different training data selection strategies. Specifi-

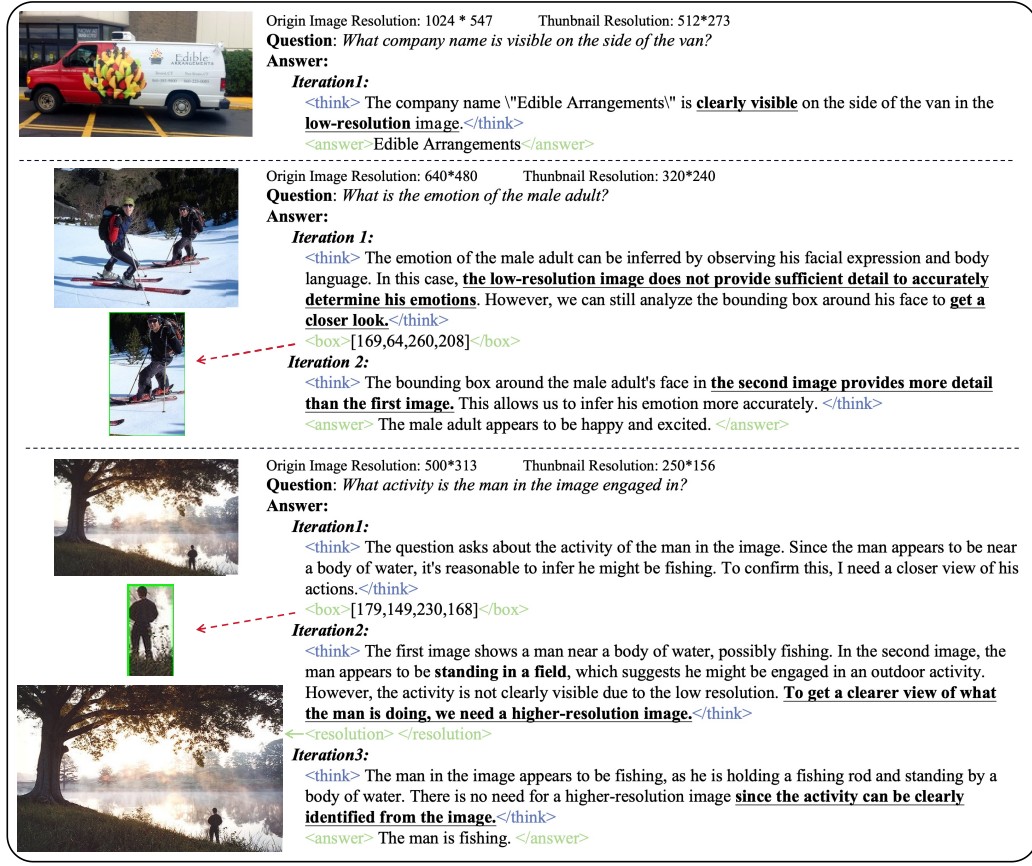

Origin Image Resolution: 1024 * 547          Thunbnail Resolution: 512*273
**Question**: *What company name is visible on the side of the van?*
**Answer:**
> *Iteration1:*
>> <think> The company name \"Edible Arrangements\" is **clearly visible** on the side of the van in the **low-resolution** image.</think>
>> <answer>Edible Arrangements </answer>

Origin Image Resolution: 640*480          Thunbnail Resolution: 320*240
**Question**: *What is the emotion of the male adult?*
**Answer:**
> *Iteration 1:*
>> <think> The emotion of the male adult can be inferred by observing his facial expression and body language. In this case, **the low-resolution image does not provide sufficient detail to accurately determine his emotions**. However, we can still analyze the bounding box around his face to **get a closer look.**</think>
>> <box>[169,64,260,208]</box>
> *Iteration 2:*
>> <think> The bounding box around the male adult's face in **the second image provides more detail than the first image.** This allows us to infer his emotion more accurately. </think>
>> <answer> The male adult appears to be happy and excited. </answer>

Origin Image Resolution: 500*313          Thunbnail Resolution: 250*156
**Question**: *What activity is the man in the image engaged in?*
**Answer:**
> *Iteration1:*
>> <think> The question asks about the activity of the man in the image. Since the man appears to be near a body of water, it's reasonable to infer he might be fishing. To confirm this, I need a closer view of his actions.</think>
>> <box>[179,149,230,168]</box>
> *Iteration2:*
>> <think> The first image shows a man near a body of water, possibly fishing. In the second image, the man appears to be **standing in a field**, which suggests he might be engaged in an outdoor activity. However, the activity is not clearly visible due to the low resolution. **To get a clearer view of what the man is doing, we need a higher-resolution image.**</think>
>> <resolution> </resolution>
> *Iteration3:*
>> <think> The man in the image appears to be fishing, as he is holding a fishing rod and standing by a body of water. There is no need for a higher-resolution image **since the activity can be clearly identified from the image.**</think>
>> <answer> The man is fishing. </answer>

*Figure 5.* Case study. Three examples show that, with RTAF, the model uses low-res thumbnails for reasoning and high-res crops for precise answering, like identifying company, emotion, and activity, balancing efficiency and effectiveness.

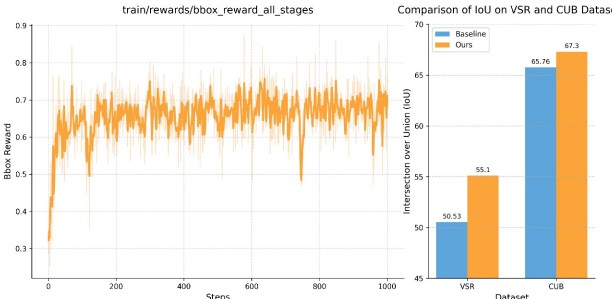

*Figure 6.* Left: The training curve of the bounding box reward. Right: The comparison of the Intersection-over-Union metric on the VSR (Liu et al., 2023) and CUB (Wah et al., 2011) benchmarks.

cally, we first use a random selection strategy (as adopted in previous works), which yields an average V Token of 553 and an accuracy of 77.1. This serves as a baseline, demonstrating the potential for improvement through deliberate data selection. Furthermore, we evaluate different data combinations of our proposed data selection strategy. Combination 1 (Class1(2k) + Class2(3k) + Class3(3k)) reduces the average V Token to 551 while boosting accuracy

to 77.6. Combination 2 (Class1(2k) + Class2(4k) + Class3 (2k)) shows limited difference, with an average V Token of 513 and an accuracy of 77.6. Notably, Combination 3 (Class1(2k) + Class2(6k)) achieves the best performance, with the average V Token reduced to 471 and an accuracy of 78.3. These results underscore that our data selection strategy, through tailored class-wise data combinations, effectively reduces computational consumption (via fewer visual tokens) while improving model accuracy, validating its role in optimizing grounded visual reasoning.

**Analysis of the grounding ability.** To verify the grounding ability of our proposed method, in Figure 6 (Left), we first report the training curve of the bounding box reward. The accuracy of evidence regions generated by our model improves steadily over the course of training, which demonstrates the effectiveness of our training strategy.

Additionally, we provide the comparison of the Intersection-over-Union metric on the VSR (Liu et al., 2023) and the CUB (Wah et al., 2011) benchmarks, as shown in Figure 6 (Right). In these two benchmarks, our model consistently outperforms the baseline model (Cao et al., 2025), i.e., an improvement of 4.57% in VSR and 1.54% in CUB, which

is attributed to our proposed training strategy, including the reward functions and the training data selection strategy.

### 4.5. Case Study

To validate the effectiveness and efficiency of the proposed RTAF paradigm, we select several cases involving three distinct examples (as presented in Figure 5), each showcasing different question types and image attributes to thoroughly assess how RTAF utilizes low-resolution and high-resolution images for precise answering. In the first example, to identify the company name on a van, the low-resolution suffices for directly spotting "Edible Arrangements", avoiding full high-resolution processing. The second example involves inferring a male adult's emotion, which indeed requires a fine-grained understanding of the man's face and body. The low-resolution thumbnail first helps reason about the region; a high-resolution crop then enables accurate detection of "happy and excited" emotion. For the third example, identifying a man's activity, the thumbnail lets the model infer "possible fishing" via background water, but details are unclear. The output key region bounding box is too small to judge further. Then, a higher-resolution image is used—revealing the man's fishing rod, confirming the correct answer "fishing". It is worth noting that in this example, the number of visual tokens used is slightly higher than that of conventional non-GVR methods. However, this additional consumption is justified as it guarantees the correctness of the answer, and such cases account for only a small proportion. Overall, these examples show RTAF efficiently uses low-resolution thumbnails for region reasoning and high-resolution crops for accurate answering, cutting computation while ensuring effectiveness.

### 5. Conclusion and Discussion

In this paper, we propose the RTAF paradigm for efficient and effective Grounded Visual Reasoning (GVR), using low-resolution thumbnails for region reasoning and high-resolution crops for answering. For effective training, we propose new reward functions and a training data selection strategy. Our model reduces visual token usage by half (vs. GVR methods) and achieves 5.34% performance gains (vs. non-GVR methods) across benchmarks, demonstrating a balance between interpretability, performance, and efficiency. Extensive experiments and ablations validate the effectiveness of our proposed new paradigm and model.

Regarding limitations, we have identified specific extreme cases where a key region occupies an exceptionally small area of the original image and lacks any salient reference objects to assist in localization (unlike scenarios where a small item is held by a person or a symbol is framed by a vehicle). In such situations, the thumbnail may fail to provide any structural or contextual cues whatsoever.

### Impact Statement

This paper presents work whose goal is to advance the field of machine learning. There are many potential societal consequences of our work, none of which we feel must be specifically highlighted here.

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
