# OpenReview forum: "Reason with Thumbnails, Answer with Focus: An Efficient and Effective Paradigm for Multimodal Grounded Visual Reasoning"
_ICML.cc/2026/Conference — ICML 2026 regular_

### Official Review · Reviewer_MvcD · 2026-02-26

**Soundness:** 2
**Presentation:** 3
**Significance:** 3
**Originality:** 3
**Overall Recommendation:** 4
**Confidence:** 3

**Summary:**

This paper proposes RTAF (Reason with Thumbnails, Answer with Focus), a grounded visual reasoning paradigm for multimodal LLMs that first reasons over low-resolution thumbnails to identify task-relevant regions, then answers using high-resolution crops from the original image; the model can optionally request a higher-resolution image when thumbnails are insufficient. The authors train purely with RL using verifiable rewards and introduce a resolution-aware training data selection pipeline. On VisCoT and RefCOCO/+, RTAF achieves higher accuracy than strong baselines while using substantially fewer visual tokens than prior GVR methods, with thorough ablations indicating each component’s contribution.

**Compliance With Llm Reviewing Policy:**

Affirmed.

**Final Justification:**

This paper is technically solid and has a certain degree of originality. In the rebuttal and subsequent responses, the authors addressed most of my concerns. Therefore, my final recommendation is 4 (weak accept).

**Key Questions For Authors:**

- How does the magnitude/weight of the efficiency reward affect the model’s behavior? In particular, does a larger efficiency reward make the model overly cautious about using tools, thereby hurting performance on complex problems that genuinely require tool calls?

- How are the ground-truth bounding boxes for the bbox reward obtained? Which datasets/tasks provide box annotations, and what is the procedure when a task does not natively include bounding-box supervision?

**Limitations:**

This paper does not include a limitations section. The authors should discuss the limitations of their reward design, and use failure cases to explore other potential limitations.

**Strengths And Weaknesses:**

## Strengths

- S1. Overall, the technical approach is reasonable: by moving from coarse to fine in reasoning and answering, it provides a multimodal grounded visual reasoning scheme that is easy to understand and efficient.

- S2. Broad evaluation across 12 VisCoT tasks and two grounding benchmarks (RefCOCO/+) demonstrates consistent improvements over the Qwen2.5‑VL base and a strong GVR baseline (Ground‑R1), with about half the visual tokens vs Ground‑R1.

- S3. Case studies concretely illustrate how RTAF balances efficiency with accuracy across easy and fine-grained questions.

## Weakness

- W1. My main concern is the efficiency reward. Its definition appears to contradict the stated property that “the model will consistently receive the same total reward whether any tool is used when the answer is correct.” As defined, this reward makes correct answers with tool usage receive a lower total reward, which can bias the model against making necessary tool calls. The authors should clearly discuss how the efficiency reward affects the model’s behavior.

- W2. The bbox reward relies on ground-truth boxes, but it is unclear which datasets provide gold boxes for all tasks used in RL and ablations; many VisCoT subtasks do not natively include bounding-box supervision.

- W3. It would also help to include some failure cases and a limitations discussion.

W1 and W2 significantly weaken the paper’s soundness in my view. Overall, I’m on the borderline, and my final score will largely depend on how the authors address these concerns in the rebuttal.

---

> ### Author Rebuttal · Authors · 2026-03-31
>
> $\textbf{Dear Reviewer MvcD,}$
>
> We sincerely thank you for your positive assessment of our work, particularly for recognizing RTAF as a reasonable, efficient, and well-evaluated grounded visual reasoning paradigm. We deeply appreciate your insightful questions regarding our reward design and supervision signals, which we address below.
>
> **Q1: Does the efficiency reward make the model overly cautious about using tools?**
>
> Response: This is a highly insightful question, and it touches upon a core consideration in our reward design.
> To prevent the model from becoming overly cautious, **the efficiency reward is explicitly designed to be counterbalanced by the grounding/bbox reward**.
>
> When a question genuinely requires zooming in, the model foregoes the efficiency reward but earns the bounding box reward by successfully locating the key region via tool usage. We specifically calibrated these two rewards to be approximately equal in magnitude.
> As a result, the total reward for correctly answering a complex question with the tool is fundamentally equivalent to answering the question without the tool.
>
> However, your intuition about the weight of the efficiency reward is absolutely correct. During our hyperparameter tuning, we observed that **if we add a weight to the efficiency, the model indeed become overly cautious and attempting to answer directly from the thumbnail**, which hurts performance on complex problems.
> We will add a detailed discussion in the revised manuscript.
>
> **Q2: Ground-truth bounding boxes source for the bbox reward.**
>
> Response: To compute the bbox reward during RL training, we only utilized subsets of the training data that natively contain ground-truth bounding box annotations.
>
> **Q3: Limitations and Failure Cases.**
>
> Response:
> Regarding limitations, we have identified specific extreme cases where **a key region occupies an exceptionally small area of the original image and lacks any salient reference objects to assist in localization (unlike scenarios where a small item is held by a person or a symbol is framed by a vehicle)**. In such situations, the thumbnail may fail to provide any structural or contextual cues whatsoever.
>
> In such scenarios, the model cannot effectively narrow down its focus and may be forced to perform an exhaustive search by requesting multiple high-resolution crops.
> Consequently, for these specific edge cases, our method might actually consume more visual tokens than other methods, or even fail to return a valid answer. We will transparently discuss this limitation in the final version.

---

> > ### Author Rebuttal · Reviewer_MvcD · 2026-04-01
> >
> > Thank you to the authors for the rebuttal.
> >
> > The response to my first question helped clarify the intended role of the efficiency reward. In particular, the explanation that the efficiency reward is designed to be counterbalanced by the bbox reward makes the design intuition clearer, and I appreciate the authors’ acknowledgement that overweighting the efficiency reward can indeed make the model overly cautious about tool use. That said, this point is still not fully resolved for me, because the rebuttal does not provide empirical evidence demonstrating that the proposed calibration is sufficient in practice.
> >
> > For my second question, I do not think the rebuttal adequately addresses the concern. The statement that bbox rewards are computed only on subsets with native ground-truth box annotations is helpful at a very high level, but it does not provide the concrete information I was hoping to see, such as which public datasets are actually used for training, how much data comes from each source, and how the final RL training set is composed. As a result, the soundness concern related to supervision and training data provenance remains.
> >
> > I appreciate the authors’ added discussion of limitations and failure cases, which improves the transparency of the work.
> >
> > Overall, while the rebuttal provides useful clarification, it does not substantially change my assessment of the paper. I will therefore keep my original scores for now and look at the opinions of the other reviewers before making any further adjustment.

---

> > > ### Author Response · Authors · 2026-04-01
> > >
> > > Thank you for your **prompt and constructive follow-up**. We deeply appreciate your meticulousness; we agree that providing concrete empirical evidence and exact data composition is essential for the paper's soundness. We sincerely apologize for the lack of specificity in our previous response, and we provide the detailed factual data below.
> > >
> > > **Q1. Empirical Evidence for Efficiency Reward Calibration**
> > >
> > > To empirically demonstrate that our calibrated efficiency reward ($R_{eff}$) does not make the model overly cautious in practice, **we add a row, which uses 2 x** $R_{eff}$ **, in the ablation study on the reward weights** (Table 5 in the manuscript).
> > > As shown in the table below, we tracked Average Visual Tokens, Accuracy (ACC), and Average Trajectory Length (which acts as a proxy for tool usage, where shorter trajectories indicate reluctance to use tools).
> > >
> > > | Rewards            | AVG V Token ↓ | ACC↑  | AVG Trajectory length|
> > > |:-|:-:|:-:|:-:|
> > > | $R_{format}$ + $R_{acc}$ | 627              | 76.2  | 3.2 |
> > > | $R_{format}$ + $R_{acc}$ + $R_{bbox}$       | 592              | 76.7  | 3.0 |
> > > | $R_{format}$ + $R_{acc}$ + $R_{bbox}$ + $R_{eff}$        | 553              | 77.1  | 2.5 |
> > > | $R_{format}$ + $R_{acc}$ + $R_{bbox}$+ **2 x** $R_{eff}$       | 282              | 74.5  | 1.4 |
> > >
> > > When the efficiency reward is set to our proposed calibrated weight (row 3), the model achieves the highest accuracy (77.1%) with a healthy average trajectory length of 2.5, proving it actively and willingly uses the tool for necessary visual exploration.
> > > However, **when we overweight the efficiency reward (row 4, 2 x $R_{eff}$), the trajectory length drastically drops to 1.4**. This empirically proves that: an overweighted efficiency reward indeed makes the model "overly cautious," forcing it to answer directly from the thumbnail, which consequently hurts performance.
> > > These results demonstrate that the proposed calibration is sufficient in practice.
> > >
> > > **2. Concrete Composition of the RL Training Set**
> > >
> > > Regarding data provenance, we guarantee that no pseudo-labels or external detectors were used. **ALL data in the VisualCoT dataset is annotated with intermediate bounding boxes highlighting key regions essential for answering the questions.**
> > > After our resolution-aware data selection pipeline, the final RL training set consists of 8K samples in total.
> > > Specifically, the training data are composed of 12 datasets from VisualCoT, with the sample count of each dataset listed as follows:
> > >
> > > | Dataset | Sample count |
> > > | :--- | :---: |
> > > | flickr30k | 2784 |
> > > | infographicsvqa | 277 |
> > > | openimages | 797 |
> > > | gqa | 2069 |
> > > | docvqa | 386 |
> > > | textcap | 417 |
> > > | dude | 181 |
> > > | textvqa | 238 |
> > > | v7w | 657 |
> > > | sroie | 65 |
> > > | cub | 87 |
> > > | vsr | 42 |

---

### Official Review · Reviewer_1ekt · 2026-03-07

**Soundness:** 3
**Presentation:** 3
**Significance:** 3
**Originality:** 2
**Overall Recommendation:** 3
**Confidence:** 3

**Summary:**

The work analyzes the problem of efficient and effective grounded visual reasoning for MLLMs. The paper proposes a novel two-stage paradigm called RTAF, in which the model first reasons over low-resolution thumbnails to identify key image regions and then uses high-resolution crops to produce the final answer. A suite of reward functions and a resolution-aware training data selection strategy are introduced to guide the GRPO training. Experiments across a wide range of benchmarks demonstrate that RTAF reduces visual token usage while achieving higher or comparable accuracy. Ablation studies show the contributions of the RTAF paradigm, reward functions, and data selection in balancing efficiency, performance, and interpretability.

**Compliance With Llm Reviewing Policy:**

Affirmed.

**Final Justification:**

I appreciate the authors’ clarifications regarding my concerns. However, I am still not fully convinced of the novelty of the RTAF framework compared with other tool-based methods, and the performance gains on HR 8K remain limited. I also note that other reviewers hold a more positive view of the novelty and overall technical solidity of this work. Therefore, I will lower my confidence.

**Key Questions For Authors:**

1. Could you provide a more detailed analysis of multi-round reasoning complexity, specifically the effect of the number of thumbnail and focus iterations on efficiency and accuracy?
2. Please clarify comparisons with thinking-with-image methods, particularly regarding the use of crop-image tools for key region identification.

**Limitations:**

1. Limited discussion of complex scenarios, such as extreme image resolutions or very small target regions.
2. Potential contradiction between multi-turn reasoning trajectories and reduced visual token usage, which may affect computational efficiency and overall performance.

**Strengths And Weaknesses:**

Strengths
1. The paper introduces a two-stage reasoning paradigm that simulates human visual perception and reasoning processes, balancing both efficiency and performance.
2. Extensive evaluation across diverse benchmarks demonstrates the robustness of the proposed method, and detailed ablation studies analyze the impact of training data and reward functions.
3. The paper presents a comprehensive training and reward design, including a suite of reward functions that effectively enhance the reinforcement learning training stage.


Weakness
1. Comparison with thinking-with-image methods is absent, in which the crop-image tool can be used to identify the key image region for question answering.
2. The authors should consider high-resolution benchmarks that focus on fine-grained understanding (e.g., HR-bench), and compare with other crop-image-based models. The reported performance gain over the baseline Qwen2.5-VL-7B appears limited.
3.  The length of the reasoning trajectory is unclear. Although visual token usage is reduced, longer trajectories may offset efficiency gains.
4. The impact of the number of thumbnail and focus rounds on efficiency and accuracy is not deeply analyzed.
5.  It is unclear whether the training data selection pipeline aligns with the reward design, i.e., whether different reward functions are applied to different types of training data.
6. The interactions among reward components and hyperparameter sensitivity are not fully clarified, including robustness to weight variations.

---

> ### Author Rebuttal · Authors · 2026-03-31
>
> $\textbf{Dear Reviewer 1ekt,}$
>
> We sincerely thank you for acknowledging our comprehensive training design, robust evaluation, and detailed ablation studies. We appreciate your constructive feedback, which has helped us clarify the efficiency trade-offs and hyperparameter settings of our approach. Below we address your specific concerns:
>
> **Q1: Comparison with "thinking-with-image" methods and performance on high-resolution benchmarks.**
>
> Response: Thank you for this suggestion. To directly address your concern, we have expanded our evaluation to include high-resolution benchmarks (i,e., HR-4K and HR-8K) and compared RTAF with recent "thinking-with-image" (crop-image-based) methods, such as DeepEyes, and Pixel-Reasoner.
>
> | Method         | AVG V Token (HR 4K) ↓ | ACC (HR 4K) ↑|  AVG V Token (HR 8K) ↓| ACC (HR 8K) ↑|
> |-|:-:|:-:|:-:|:-:|
> | **Ours**           | **13118** | **73.8**            | **26772**                   | **69.5**            |
> | Qwen2.5VL-7B   | 15829| 68.2            | 16269                   | 62.7            |
> | DeepEyes       | 41080  | 73.2            | 30999                   | 69.5            |
> | Pixel Reasoner | 39418  | 74.0            | 35856                       | 66.9            |
>
> The results demonstrate that RTAF not only remains highly competitive in fine-grained understanding but also operates with substantially higher efficiency. compared the baseline Qwen2.5-VL-7B, our method achieve 5.6 and 6.8 performance gains in the HR 4k and 8K, 5.8 in the VisCoT, this is not limited performance gains.
>
> **Q2: Length of reasoning trajectory and multi-round reasoning complexity.**
>
> Response: It is a very insightful question regarding whether longer text reasoning trajectories might offset the efficiency gains from reduced visual tokens.
> Empirically, our method actually shortens the reasoning trajectory compared to previous GVR methods. **RTAF requires an average of only 2.4 rounds of interaction, compared to 3.14 rounds for Ground-R1 in the VisCoT benchmark**. This reduction is primarily attributed to our proposed efficiency reward, which penalizes overly verbose exploration and encourages the model to locate key regions decisively.
> Furthermore, because RTAF only processes thumbnails and small local crops, the overall computational cost (and latency) is significantly reduced in multi-turn scenarios.
>
> **Q3: Alignment of training data selection and reward design.**
>
> Response: To clarify, the training data selection pipeline and the reward design are two orthogonal but highly synergistic processes aimed at efficient training.
> **We apply the same suite of reward functions uniformly across all selected training data.** The data selection pipeline (filtering for "resolution-sensitive" and "resolution-neutral" samples) ensures that the model is exposed to instances where zooming in is actually necessary and solvable—preventing it from wasting capacity on trivial or impossibly hard samples. Meanwhile, the uniform reward functions teach the model how to execute the task (optimizing for format, accuracy, localization, and efficiency). They work together to accelerate convergence without needing data-specific reward rules.
>
> **Q4: Interactions among reward components and hyperparameter sensitivity.**
>
> Response: We conducte hyperparameter sensitivity analyses to explore the interplay between the reward components. Specifically, we focused on the accuracy and efficiency rewards:
> 1. Accuracy Reward: We experimented with scaling up the weight of the accuracy reward. We found that **increasing its weight has a marginal impact on the final performance**. The base reward signal, when combined with the format and grounding rewards, is already sufficient to effectively guide the model toward correct answers without requiring heavy tuning.
> 2. Efficiency Reward: During our experiments, we observed that **if we add a higher weight to the efficiency penalty, the model actually becomes overly cautious and attempts to answer directly from the thumbnail**. This reluctance to utilize the focus tool significantly hurts its performance on complex problems that genuinely require fine-grained visual details.
>
> **Q5: Limitations regarding complex scenarios (extreme resolutions/tiny targets).**
>
> Response: We agree with your observation. We have identified specific extreme cases where a key region occupies an exceptionally small area of the original image and lacks any salient reference objects to assist in localization (unlike scenarios where a small item is held by a person or a symbol is framed by a vehicle). In such situations, the thumbnail may fail to provide any structural or contextual cues whatsoever. This forces the model into an exhaustive spatial search, which inevitably increases visual token usage. We will include a detailed discussion of this edge case in our newly added Limitations section.

---

> > ### Author Rebuttal · Reviewer_1ekt · 2026-04-03
> >
> > Thank you for your response and clarifications. However, the distinctions between the proposed method and existing thinking-with-images approaches are not sufficiently clarified. Concerns also remain regarding the robustness of the method in complex scenarios. Furthermore, the performance improvements over competing methods are relatively modest on high-resolution benchmarks  such as  HR-Bench. I will therefore maintain my original score.

---

> > > ### Author Response · Authors · 2026-04-05
> > >
> > > **RE1:   Significant Improvement in the Accuracy-Efficiency Trade-off:**
> > >
> > > **We respectfully disagree that the improvements are marginal.** If we look strictly at the efficiency-performance joint metrics, RTAF demonstrates a significant advantage. At HR 4K, RTAF matches the SOTA accuracy of Pixel-Reasoner (73.8 vs 74.0) while consuming only 33% of the visual tokens (13,118 vs 39,418). **An efficiency gain of this magnitude while preserving fine-grained reasoning accuracy represents a substantial contribution to the community.**
> > >
> > > **RE2:  Distinct and Novel Technical Paradigm:**
> > >
> > > Regarding novelty, grouping RTAF with existing GVR methods overlooks our core technical innovation.
> > > While our approach shares the high-level concept of a 'zoom-in' (cropping) operation with prior GVR methods, the similarities end entirely there. **Driven by the critical need to achieve vastly superior efficiency without sacrificing performance, we introduce RTAF**. RTAF represents a fundamental paradigm shift: it inherently empowers the model to perform preliminary spatial reasoning using low-resolution thumbnails, selectively reserving high-resolution inputs strictly for the final answering phase. **This coarse-to-fine decoupling is fundamentally distinct from previous exhaustive methodologies**. Crucially, across a diverse array of benchmarks, our empirical results validate our core design objective—RTAF consistently delivers superior or highly competitive performance while consuming only a fraction of the visual tokens.

---

### Official Review · Reviewer_J48r · 2026-03-12

**Soundness:** 2
**Presentation:** 2
**Significance:** 2
**Originality:** 2
**Overall Recommendation:** 3
**Confidence:** 4

**Summary:**

The authors refine the existing 'ground-then-answer' reasoning paradigm by introducing a novel approach called Reason with Thumbnails, Answer with Focus (RTAF). In this paradigm, the model leverages low-resolution images to reason about key areas and utilizes high-resolution crops for final answer generation. The method incorporates reinforcement learning equipped with a suite of reward functions to guide model behavior, alongside a resolution-aware training data selection strategy. Extensive experiments demonstrate that these designs yield improved performance while reducing computational costs.

**Compliance With Llm Reviewing Policy:**

Affirmed.

**Final Justification:**

I appreciate the authors’ detailed response and acknowledge that the proposed coarse-to-fine method is methodologically sound and empirically effective. Accordingly, I have increased my rating by one point. However I retain concerns regarding the work’s incremental novelty, as the approach primarily integrates established techniques.

**Key Questions For Authors:**

Please refer to the weaknesses part.

**Limitations:**

This paper lacks a discussion about the limitations and social impact of the proposed method.

**Strengths And Weaknesses:**

Strengths:

1.	The proposed "Reason with Thumbnails, Answer with Focus" (RTAF) paradigm is well-motivated and technically sound.
2.	The paper is well-organized, and easy to follow.

Weaknesses:

1.	The technical novelty of this work is somewhat incremental. The core strategy of utilizing thumbnails for reasoning offering limited distinction from prior Grounded Visual Reasoning methods.
2.	The use of thumbnails may lead to information loss, making it difficult to recover fine-grained details. This is particularly problematic for high-resolution benchmarks like V*, HR-Bench, and MME-RealWorld, which usually inquire about subtle objects in high-resolution images,
3.	Including more high-resolution benchmarks, such as V*, HR-Bench, or MME-RealWorld, would strengthen the evaluation.
4.	The paper lacks comparison with recent Grounded Visual Reasoning (GVR) methods, e.g., DeepEyes [1], Thyme[2] and Pixel-Reasoner[3].

[1] Zheng Z, Yang M, Hong J, et al. Deepeyes: Incentivizing" thinking with images" via reinforcement learning.

[2] Zhang Y F, Lu X, Yin S, et al. Thyme: Think beyond images.

[3] Wang H, Su A, Ren W, et al. Pixel reasoner: Incentivizing pixel-space reasoning with curiosity-driven reinforcement learning.

---

> ### Author Rebuttal · Authors · 2026-03-31
>
> $\textbf{Dear Reviewer J48r,}$
>
> We sincerely appreciate your detailed feedback and for recognizing our work as well-motivated, technically sound, and well-organized. We understand your concerns regarding technical novelty and potential information loss, and we believe the following clarifications and new baseline evaluations will alleviate your concerns.
>
> **Q1: Incremental novelty and comparison with recent GVR methods (DeepEyes, Pixel-Reasoner) in high-resolution benchmarks.**
>
> Response: While RTAF shares the high-level motivation of "zooming in" with recent Grounded Visual Reasoning (GVR) methods, our core technical contribution fundamentally differs in how it resolves the immense computational bottlenecks of existing paradigms. Specifically:
> 1. Methodology: Our approach fundamentally restructures the visual reasoning process into a "coarse-to-fine" paradigm. Instead of encoding the entire high-resolution image upfront, which incurs massive computational costs, **our approach utilizes low-resolution thumbnails to establish global context and perform initial spatial reasoning**, which does not necessaryly require fine-grained perception in most cases. The model then dynamically and selectively requests high-resolution crops ("Focus") only for regions requiring fine-grained inspection. This explicit spatial decoupling drastically reduces redundant visual token usage while strictly preserving critical local details.
> 2. Training Strategy: Unlike previous methods that rely heavily on complex heuristics or multi-stage pipelines (e.g., Pixel-Reasoner uses Warm-Start Instruction Tuning and a Curiosity-Driven Reinforcement Learning), our approach adopts a **pure RL (GRPO)** framework paired with **a suite of bespoke reward** functions that jointly optimize for format, accuracy, localization, and efficiency. Additionally, We introduce a novel **resolution-aware training data selection strategy**, which is critical for preventing the model from overfitting to trivial or overly difficult samples, with high efficiency.
>
> To directly address your concern, we compare our method with DeepEyes, and Pixel-Reasoner on the HR-Bench. As shown below, RTAF achieves a dual advantage: it maintains highly competitive performance while significantly reducing computational costs (e.g., drastically reducing average visual tokens).
>
> | Method         | AVG V Token (HR 4K) ↓ | ACC (HR 4K) ↑|  AVG V Token (HR 8K) ↓| ACC (HR 8K) ↑|
> |-|:-:|:-:|:-:|:-:|
> | **Ours**           | **13118** | **73.8**            | **26772**                   | **69.5**            |
> | Qwen2.5VL-7B   | 15829| 68.2            | 16269                   | 62.7            |
> | DeepEyes       | 41080  | 73.2            | 30999                   | 69.5            |
> | Pixel Reasoner | 39418  | 74.0            | 35856                       | 66.9            |
>
>
> Due to the rebuttal time limitation, we include the HR bench which evaluate the performance on 4K and 8K high-resolution images, we will incorporate more benchmarks in the final version.
>
> **Q2: Information loss on high-resolution benchmarks.**
>
> Response: We completely agree that identifying subtle objects in high-resolution images is challenging.
> However, we would like to highlight a crucial observation regarding visual reasoning: **locating the key region typically does not require fine-grained perception from the very beginning**. In most practical scenarios, the visual question provides strong macroscopic anchors. For instance, to answer questions about "the object in a person's hand" or "the sign on a car," the model initially only needs to locate the "person" or the "car." These primary entities are highly salient and can be easily identified using only the low-resolution thumbnail.
>
> Additionally, we explicitly designed RTAF to alleviate irreversible information loss. In our paradigm, **the thumbnail serves as a structural overview rather than the sole source of information** . When the thumbnail lacks sufficient fine-grained details (which would otherwise lead to blind guessing), the model is trained to actively utilize a tool to request high-resolution crops of specific suspicious regions.Also, the results on the HR bench also demonstrate the effective of our method.

---

> > ### Author Rebuttal · Reviewer_J48r · 2026-04-04
> >
> > Thank you for the authors' response. However, the comparison with existing Grounded Visual Reasoning (GVR) methods on HRBench demonstrates only marginal improvements, and my concerns regarding the incremental novelty of this work remain unresolved. Therefore, I have decided to maintain my original score.

---

> > > ### Author Response · Authors · 2026-04-05
> > >
> > > **RE1:   Significant Improvement in the Accuracy-Efficiency Trade-off:**
> > >
> > > **We respectfully disagree that the improvements are marginal.** If we look strictly at the efficiency-performance joint metrics, RTAF demonstrates a significant advantage. At HR 4K, RTAF matches the SOTA accuracy of Pixel-Reasoner (73.8 vs 74.0) while consuming only 33% of the visual tokens (13,118 vs 39,418). **An efficiency gain of this magnitude while preserving fine-grained reasoning accuracy represents a substantial contribution to the community.**
> > >
> > > **RE2:  Distinct and Novel Technical Paradigm:**
> > >
> > > Regarding novelty, grouping RTAF with existing GVR methods overlooks our core technical innovation.
> > > While our approach shares the high-level concept of a 'zoom-in' (cropping) operation with prior GVR methods, the similarities end entirely there. **Driven by the critical need to achieve vastly superior efficiency without sacrificing performance, we introduce RTAF**. RTAF represents a fundamental paradigm shift: it inherently empowers the model to perform preliminary spatial reasoning using low-resolution thumbnails, selectively reserving high-resolution inputs strictly for the final answering phase. **This coarse-to-fine decoupling is fundamentally distinct from previous exhaustive methodologies**. Crucially, across a diverse array of benchmarks, our empirical results validate our core design objective—RTAF consistently delivers superior or highly competitive performance while consuming only a fraction of the visual tokens.

---

### Official Review · Reviewer_vR4D · 2026-03-13

**Soundness:** 3
**Presentation:** 3
**Significance:** 3
**Originality:** 2
**Overall Recommendation:** 4
**Confidence:** 3

**Summary:**

The paper approaches the problem of reasoning about visual content by zooming into certain parts of the visual content and providing an answer or result. The core question this paper addresses is how to increase efficiency by only using a low-resolution input to start with.

To approach this problem, the paper proposes a new approach, Reason with Thumbnails, Answer with Focus (RTAF) which is trained via GRPO. The reward for GRPO includes the model's efficiency besides the format of the ouptut, accuracy, and the correct localization. The model is trained with data, where zooming in helps or resolution does not change results. However, samples that are either always correct or always incorrect when zooming in or not zooming in are discarded and not used for training as they are too easy or too difficult, respectively. The paper shows that this helps with improving learning and performance.

**Compliance With Llm Reviewing Policy:**

Affirmed.

**Final Justification:**

I think the authors addressed my questions, especially the discussion and comparison to prior work, and the additional comparison on the HR benchmark. The author shows that their approach is significantly more efficient w.r.t. the number of tokens compared to prior work.

I keep my recommendation with weak accept; I think the paper can be accepted assuming the authors make the changes discussed in the rebutal; that said they are significant changes, given that the authors will have to adjust their claims a bit to move way from the main characterization of their work as "which feeds the model with low-resolution images to reason the relevant regions and high-resolution crops to generate the final answer." to characterize it as in the rebutal to delinate it better to prior work (which were unfortunatly not discussed in the original submission).

**Key Questions For Authors:**

1. The paper builds partially on prior work Cao et al., 2025. I am wondering why the authors do not compare on the benchmarks reported in Cao et al., 2025, besides RefCOCO.
2. The paper works with about 8000 training samples. How is the effect of increasing or decreasing this significantly, i.e., 0.5x, 2x, 4x, ...
3. A core contribution is the efficiency aspect. What is the AVG V Token for Table 2, also in comparison to the two other reported models in the table? Or why can they not be reported?
4. How does Ground-R1 perform without the added bbox reward?

**Limitations:**

The paper neither discussed limitations nor the potential negative societal impact of their work.

**Strengths And Weaknesses:**

## Originality
The paper introduces, to my knowledge, the novel concepts a reward for reasoning efficiency and proposes a resolution-aware training data selection strategy for effective training, by discarding some of the data.

However, there are several prior works that consider zooming in; although they do not frame it the same way, it also starts with a lower resolution image and zoom into a higher resolution image: Hu 2024, Su 2025. I am wondering why these are not discussed and compared to.
1. I think it would be great to discuss the relationship to them and also the benchmarks used in these works.
2. e.g., [Su 2025] uses a very similar idea of zooming in, and this should also be more efficient than looking at the full image; the rewards are a bit different, but there is a lot of similarity.

## Presentation
The presentation is overall clear, some minor comments:
### minor
1. The Combinations in Table 4 could have been shown what data they combine directly in the table, i.e., a column for Class 1: Resolution Sensitive, Class 2: Resolution-Neutral, Class 3: always correct or always incorrect in both resolutions.
2. The paper writes "Compared with previous non-GVR methods, our method achieves a significant improvement, i.e., +30.5% average accuracy compared with LLaVA-1.5-13B. Additionally, compared with Qwen2.5-VL-7B, our model still achieves +5.8%". The first sentence is a bit misleading/confusing. Is Qwen2.5-VL-7B as reported in the table not also a non-GVR method?




## Significance
The paper addresses the very general problem of visual question answering which can encompass a large variety of vision and lanuage tasks and the paper also evaluates across a varity of types of tasks and datasets.

## Soundness:
The paper is overall sound, with claims experimentally validated with ablation studies in table 3, 4, and 5, demonstrating the benefit of the proposed data selection, model components, and reward decomposition.

## References
* [Hu 2024] Hu, Y., Shi, W., Fu, X., Roth, D., Ostendorf, M., Zettlemoyer, L., Smith, N.A. and Krishna, R., 2024. Visual sketchpad: Sketching as a visual chain of thought for multimodal language models. Advances in Neural Information Processing Systems, 37, pp.139348-139379.
* [Su 2025] Su, A., Wang, H., Ren, W., Lin, F. and Chen, W., 2025. Pixel Reasoner: Incentivizing Pixel Space Reasoning via Curiosity-Driven Reinforcement Learning. In The Thirty-ninth Annual Conference on Neural Information Processing Systems.

---

> ### Author Rebuttal · Authors · 2026-03-31
>
> $\textbf{Dear Reviewer vR4D,}$
>
> We sincerely thank you for your constructive comments and for recognizing the originality and soundness of our proposed RTAF approach, as well as our data selection strategy. Below, we address your specific questions and concerns:
>
> **Q1. Missing comparisons with prior zooming-in methods and benchmarks from Cao et al., 2025.**
>
> Response: We sincerely appreciate you pointing out these highly relevant works.
>
> **Compared with other Ground Visual Reasoning methods**, while RTAF shares the high-level motivation of "zooming in" with them, our core technical contribution fundamentally differs in how it resolves the immense computational bottlenecks of existing paradigms. Specifically:
> 1. Methodology: Our approach  restructures the visual reasoning process into a "coarse-to-fine" paradigm. Instead of encoding the entire high-resolution image, which incurs massive computation, **our approach utilizes low-resolution thumbnails to establish global context and perform initial spatial reasoning**. The model then requests high-resolution crops ("Focus") only for regions requiring fine-grained inspection. This explicit spatial decoupling drastically reduces redundant visual token usage while strictly preserving critical local details.
> 2. Training Strategy: Unlike previous methods that rely heavily on complex heuristics or multi-stage pipelines (e.g., Pixel-Reasoner uses Warm-Start Instruction Tuning and a Curiosity-Driven Reinforcement Learning), our method adopts a **pure RL (GRPO)** framework paired with a suite of reward functions that jointly optimization. Additionally, We introduce a novel **resolution-aware data selection strategy** for efficient training.
>
> **Regarding the benchmarks** from Cao et al. (2025), they updated their paper on February 3, 2026 and added several new benchmarks, which occurred after the submission deadline. Therefore, our paper not include these benchmarks.
> To provide a comprehensive comparison with the zooming-in methods you mentioned, we have now evaluated RTAF on two high-resolution benchmarks (HR-Bench-4K and HR-Bench-8K) and compared it with Pixel Reasoner and DeepEyes. The results are as follows:
>
> | Method         | AVG V Token (HR 4K) ↓ | ACC (HR 4K) ↑|  AVG V Token (HR 8K) ↓| ACC (HR 8K) ↑|
> |-|:-:|:-:|:-:|:-:|
> | **Ours**           | **13118** | **73.8**            | **26772**                   | **69.5**            |
> | Qwen2.5VL-7B   | 15829| 68.2            | 16269                   | 62.7            |
> | DeepEyes       | 41080  | 73.2            | 30999                   | 69.5            |
> | Pixel Reasoner | 39418  | 74.0            | 35856                       | 66.9            |
>
> Due to the rebuttal time limitation, we include the HR bench which evaluate the performance on 4K and 8K images, we will incorporate more benchmark in the final version.
>
> **Q2: Effect of increasing/decreasing training samples (0.5x, 2x, 4x).**
>
> Response: We agree that data scaling is an important aspect. In our current setup, the 8,000 training samples represent the entirety of our available filtered data. This upper bound is primarily constrained by the scarcity of "Resolution-Sensitive" samples, which amount to only about 2,000 instances.
> While Table 4 in our paper ablates different data combinations, to directly address your question regarding data scaling, we have supplemented an ablation study using 0.25x, 0.5x and 0.75x of our training data (corresponding to 250, 500, and 750 training steps, respectively), on the VisCoT benchmark.
> | Method            | AVG V Token | ACC   |
> |-|:-:|:-:|
> | 0.25 (250 Steps)  | 682         | 76.72 |
> | 0.5 (500 Steps)   | 593         | 77.28 |
> | 0.75 (750 Steps)  | 526         | 78.70 |
> | 1 (1000 Steps)    | 471         | 78.30 |
>
> The results demonstrate how the model's performance scales with the amount of resolution-aware training data.
>
> **Q3: Clarification on the sentence regarding Qwen2.5-VL-7B and non-GVR methods.**
>
> Response: You are absolutely correct, Qwen2.5-VL-7B is indeed a non-GVR method. We will correct this sentence in the final version.
>
> **Q4: What is the Average Visual Token for Table 2, also in comparison to the two other reported models in the table? Or why can they not be reported?**
>
> Response: Table 2 evaluates the model on RefCOCO and RefCOCO+, which specifically test the model's spatial localization (grounding) capabilities. In this setting, the model is tasked with outputting the bounding box of a specified object in a single-turn interaction, so we remove the AVG V Token column.
>
> **Q5: How does Ground-R1 perform without the added bbox reward?**
>
> Response: Our experiments show that Ground-R1 (without explicitly added bbox reward) performs comparably to Ground-R1-BBox. This is consistent with the claim in the Ground-R1 paper, as it indicates that training purely via RL enables the model to accurately ground and locate the key regions, with or without the bbox supervision.

---

> > ### Author Rebuttal · Reviewer_vR4D · 2026-04-03
> >
> > I thank the reviewers for their detailed response.
> >
> > Additional comment:
> > * Unfortuantly the bolding in the provided table is misleading: Bold in the paper indiciated the highest number, but here Pixel reasoner has higher Acc (HR 4K), Qwen, better AVG Tokens, and Deep Eyes the same ACC (HR 8K)
> > * Any thoughts on why "our" in the new table increases significantly in tokens for HR 8K vs. 4k, while other models stay nearly the same or decrease?
> > * The model seems to saturate already at 75% of the training data with a rather small training dataset. This seems unusual for most ML Approaches. Any thoughts on that?

---

> > > ### Author Response · Authors · 2026-04-05
> > >
> > > We sincerely thank the reviewer for the detailed feedback and constructive observations. Please find our point-by-point responses below:
> > >
> > > 1. The bold font applied to 'Ours' in this response table was simply used as a visual separator to distinguish our proposed method from the others. We will update the table formatting in the revised final version.
> > >
> > > 2.  **Regarding the token increase at HR 8K vs. 4K:**
> > > Regarding this observation, we attribute this to the max_pixel bottleneck during inference. Other 'think with image' methods have already saturated at such high resolutions, constraining their scaling. The fact that our model scales smoothly while still using strictly fewer tokens than baseline models at 8K further highlights the token-efficiency advantage of our method.
> > >
> > > 3. **Regarding the model saturation at 75% of the training data:** We thank the reviewer for this observation. Although the accuracy (ACC) improvement diminishes after 75% of the data, we would like to emphasize that the core objective of our method is to achieve both **effectiveness and efficiency**. Notably, **the model's token efficiency continues to improve**: the average visual token count drops further from 526 at 750 steps to 471 at 1000 steps. In addition, keeping the 1000-step training schedule was necessary to maintain a fair comparison with existing methods.

---

### Decision · Program_Chairs · 2026-04-30

**Decision:**

Accept (regular)

**Comment:**

This paper ended up receiving mixed reviews after the rebuttal (2 weak accepts and 2 weak rejects).

The reasons to accept this papers are:
- The coarse-to-fine paradigm is well-motivated: starting with low-resolution thumbnails for spatial reasoning and selectively requesting high-resolution crops only when needed (Reviewer MvcD). The rebuttal's HR-Bench comparisons confirmed this yields roughly 1/3 the visual tokens of competing methods at competitive accuracy.

- The efficiency reward and resolution-aware data selection strategy are novel contributions that effectively guide the RL training (Reviewer vR4D).

- The evaluation is sound, with ablation studies demonstrating each component's contribution (Reviewers vR4D, MvcD).

The reasons to reject this paper include:
- The original submission missed discussion of closely related "thinking with images" methods, including DeepEyes, Pixel Reasoner, and Thyme (Reviewers J48r, 1ekt, vR4D).

- The accuracy improvements over these methods on high-resolution benchmarks are modest, with the gains being primarily in efficiency rather than performance (Reviewers J48r, 1ekt).

The AC found the criticism of lack of novelty not sufficient to justify rejection. The two weak-reject reviewers grouped the proposed RTAF with existing "thinking with images" methods, but the key distinction is that prior methods all start with the full-resolution image and zoom in during reasoning, consuming large numbers of visual tokens from the outset, whereas RTAF starts with low-resolution thumbnails and requests high-resolution crops only when necessary. This is a simple but meaningfully different design choice that directly addresses the computational bottleneck. The rebuttal results on HR-Bench show that RTAF uses 13K visual tokens versus 39-41K for DeepEyes and Pixel Reasoner at comparable accuracy.

Furthermore, Pixel Reasoner (NeurIPS 2025) is the only one among the missing references that was clearly published prior work at the time of the ICML deadline. DeepEyes and Thyme were arXiv preprints whose ICLR 2026 acceptances had not yet been announced publicly. Arguably, authors should not be asked to compare with them. The missing discussion is a presentation gap the authors should fix in the revision, but it does not diminish the contribution itself.

The AC thus recommends to accept this paper and highly encourages the authors to incorporate the rebuttal results into the final version.